# Differentiation of mouse fetal lung alveolar progenitors in serum-free organotypic cultures

Konstantinos Gkatzis*[†], Paolo Panza*[†], Sofia Peruzzo[‡], Didier YR Stainier*

Department of Developmental Genetics, Max Planck Institute for Heart and Lung Research, Bad Nauheim, Germany

**Abstract** Lung epithelial progenitors differentiate into alveolar type 1 (AT1) and type 2 (AT2) cells. These cells form the air-blood interface and secrete surfactant, respectively, and are essential for lung maturation and function. Current protocols to derive and culture alveolar cells do not faithfully recapitulate the architecture of the distal lung, which influences cell fate patterns in vivo. Here, we report serum-free conditions that allow for growth and differentiation of mouse distal lung epithelial progenitors. We find that Collagen I promotes the differentiation of flattened, polarized AT1 cells. Using these organoids, we performed a chemical screen to investigate WNT signaling in epithelial differentiation. We identify an association between Casein Kinase activity and maintenance of an AT2 expression signature; Casein Kinase inhibition leads to an increase in AT1/progenitor cell ratio. These organoids provide a simplified model of alveolar differentiation and constitute a scalable screening platform to identify and analyze cell differentiation mechanisms.

*For correspondence:
konstantinosgkatzis@
googlemail.com (KG);
paolo.panza@mpi-bn.mpg.de
(PP);
Didier.Stainier@mpi-bn.mpg.de
(DYRS)

[†]These authors contributed
equally to this work

Present address: [‡]Institute of
Cardiovascular Regeneration,
Goethe University, Frankfurt,
Germany

Competing interest: See page
14

Reviewing Editor: Melanie
Königshoff, University of
Pittsburgh, United States

## Introduction

Lung alveolar progenitor cells differentiate at saccular stages (E17 in mouse, 26 weeks in human) into flattened alveolar type 1 (AT1) cells, constituting the surface for gas exchange, and secretory type 2 (AT2) cells, which prevent alveolar collapse by secreting pulmonary surfactant (*Morrisey and Hogan, 2010*; *Chao et al., 2015*). Fetal distal lung progenitors are thought to be bipotent and they co-express markers of both AT1 and AT2 cells, including Podoplanin/PDPN and Advanced Glycosylation End product-specific Receptor/RAGE as well as Prosurfactant Protein C/SFTPC, respectively (*Desai et al., 2014*; *Treutlein et al., 2014*). Differentiation of progenitors into AT1 and AT2 cells is regulated by multiple signals including WNT (Wingless and Int-1) and FGF (Fibroblast Growth Factor) from the mesenchyme (*Volckaert and De Langhe, 2015*; *Li et al., 2018*), mechanical forces (*Li et al., 2018*), epigenetic modifications (*Wang et al., 2016*), and the extracellular matrix (ECM) (*Kim et al., 2018*).

In particular, WNT signaling was shown to promote the expansion of *Axin2+* AT2 cells during alveologenesis and prevent their conversion to the AT1 lineage (*Frank et al., 2016*). Similarly, WNT activity marks lung progenitors in homeostasis and regeneration, and WNT inhibition leads to AT1 differentiation (*Nabhan et al., 2018*; *Zacharias et al., 2018*). During mouse fetal development, β-catenin is required for the maintenance of distal lung progenitors (*Ostrin et al., 2018*); however, a mechanistic understanding of how WNT signaling controls AT1/AT2 cell differentiation or fate selection is currently lacking (*Aros et al., 2021*). A more detailed picture of how WNT/β-catenin signaling influences early alveolar differentiation could lead to better derivation protocols for alveolar cells and reveal new ways to stimulate differentiation in the fetal lung.

3D cultures of lung- or iPSC-derived alveolar progenitor cells have been used to model respiratory cell fate specification and signaling/cell interactions (reviewed by *Gkatzis et al., 2018*). However, these models fail to recapitulate the defined epithelial architecture of the mouse lung at canalicular and saccular stages, with HOPX+ cells residing in the stalk and SOX9+/SFTPC+ cells in the tip region

(*Frank et al., 2019*). Until recently (*Vazquez-Armendariz et al., 2020*), most models have displayed a spheroid or unpatterned morphology, making it difficult to assess localized differentiation from progenitors to alveolar cells. In addition, despite the central role of AT1 cells in lung development (*Zepp et al., 2021*), AT1 cell isolation and generation in vitro have been longstanding obstacles, and the efficiency of AT1 cell generation in vitro, or the properties of the cells obtained in these cultures, have not been analyzed in detail, limiting the relevance of these models for cell differentiation studies. These shortcomings highlight the need for an alveolar organoid model with morphological and cellular features as observed in vivo, which would accelerate the identification of differentiation-promoting factors.

To help address these questions, we developed a serum-free, rapid, and scalable distal lung progenitor organotypic culture system that recapitulates both AT1 cell differentiation and the endogenous tissue architecture of the alveolizing lung, which will facilitate mechanistic investigations of alveolar development.

## Results and discussion
### Mouse fetal alveolar epithelial progenitors are maintained in 3D serum-free cultures

To generate alveolar organoids by self-organization of endogenous progenitors, we manually dissociated E14.5 mouse lungs and collected distal epithelial fragments. At E14.5, lung pseudoglandular development is almost complete in mouse and NKX2.1$^+$ alveolar progenitors become specified (*Frank et al., 2019*), while the distal SOX9$^+$ epithelial domain begins to expand compared with the proximal airway (*Alanis et al., 2014*).

We embedded the epithelial tips into diluted Matrigel/Collagen I domes and established cultures in defined medium supplemented with growth factors (*Figure 1A*). Three main soluble growth factors have been implicated in distal lung growth and differentiation: FGF7, FGF10, and BMP4 (Bone Morphogenetic Protein 4) (*Bellusci et al., 1997*; *Weaver et al., 1999*; *Weaver et al., 2000*; *Chao et al., 2016*; *Li et al., 2018*; *Danopoulos et al., 2019*). We tested combinations of these factors and observed that treatment with FGF7 and FGF10 led to epithelial growth and lumenization over three days without inducing hyperproliferation of residual mesenchymal cells (*Figure 1—figure supplement 1A, B*). We further found that cryopreserved epithelial tips could be expanded in culture and formed organoids in FGF-supplemented medium (52% efficiency compared with 75% using freshly dissociated tissue, *Figure 1B*), thereby allowing for more experimental flexibility.

To promote progenitor differentiation in longer-term cultures, we developed a six-day culturing protocol with a one-time change of medium on day 3, when the concentration of supplemented growth factors was halved (*Figure 1A*). This reduction slowed down organoid expansion and promoted an increase in the number and density of epithelial branches. Morphologically, on day 1, the epithelial tips reorganized to form a closed system (cyst) and began a process of lumen expansion. On day 2, a single large lumen was observed, along with newly formed epithelial branches. By day 6, many epithelial branches had extended into the surrounding matrix (*Figure 1C*). Although at this stage mesenchymal cells can be observed, organoid morphogenesis did not appear to depend on their prevalence. A role for interstitial cells in local matrix remodeling and/or epithelial cell proliferation or migration, however, cannot be excluded.

To determine the location and dynamics of progenitor cells in the organoid cultures, we assessed the expression of SOX9, a transcription factor restricted to the distal progenitor population in mouse and human (*Rock et al., 2009*; *Alanis et al., 2014*; *Nikolić et al., 2017*; *Frank et al., 2019*), and which is required for lung epithelial differentiation (*Rockich et al., 2013*). SOX9$^+$ cells were prevalent in isolated structures on day 0, and SOX9 expression was maintained throughout organoid lumenization until day 2. During this process, the majority of cells in the organoids were proliferating. By day 6, the tips of newly formed epithelial branches contained fewer SOX9$^+$ cells and also fewer proliferating cells as assessed by KI67 immunostaining (*Figure 1D*). These data indicate that the distal progenitor state is initially maintained in the 3D cultures.

Next, we assessed progenitor marker mRNA levels in organoids by comparing multiple culture stages with distal epithelial tips collected at E13.5, one embryonic day prior to sample collection for organoid culture. The distal epithelial markers *Id2*, *Bmp4* and *Etv5* displayed higher expression

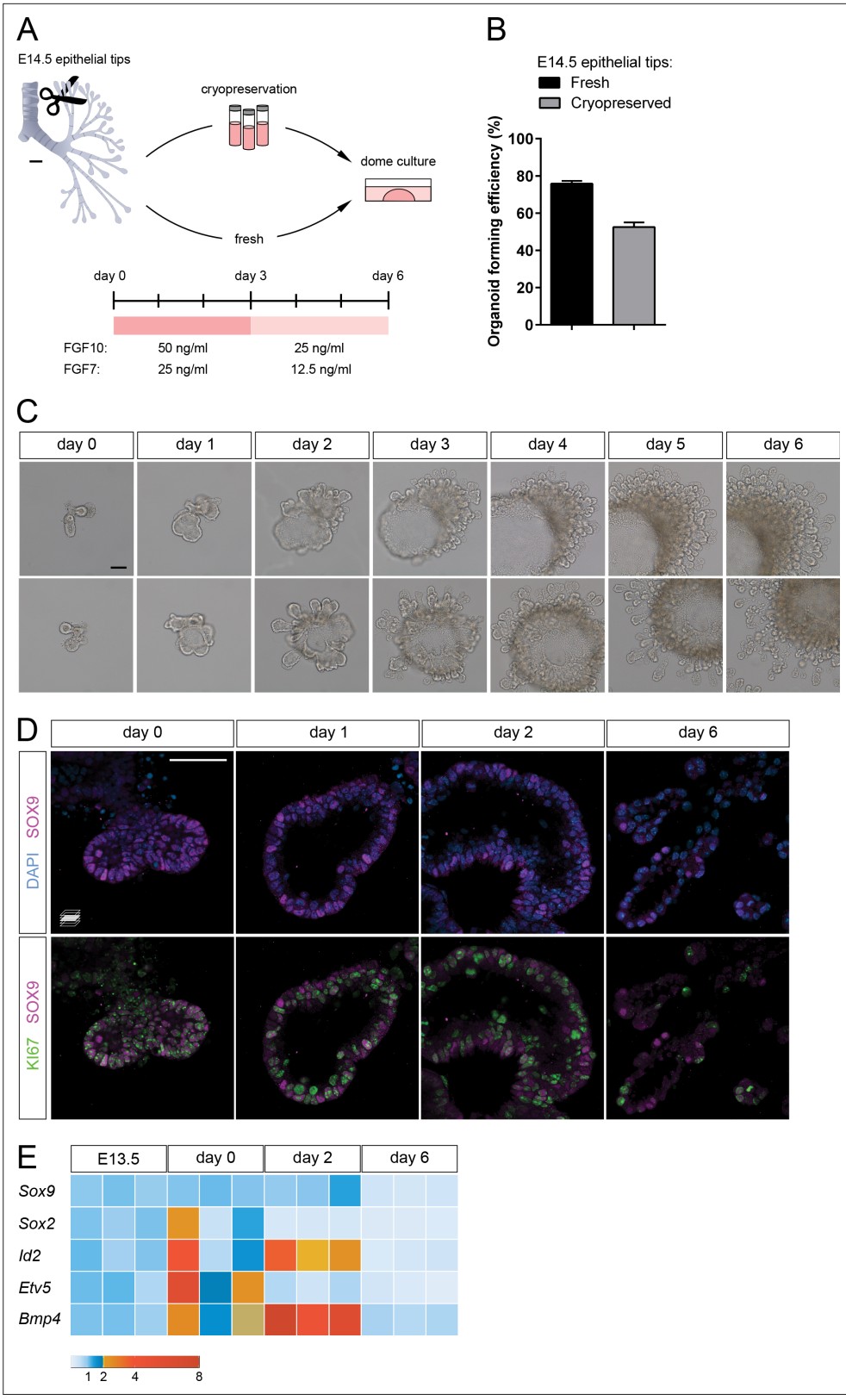

**Figure 1.** Lung distal progenitor marker expression is progressively lost in 3D explant cultures. (**A**) Schematic of the organoid establishment procedure. (**B**) Organoid forming efficiency from fresh (black bar) and cryopreserved (gray bar) epithelial tips. (**C**) Representative organoids over time (brightfield imaging). Isolated tips lumenize by day 2 and subsequently form digit-like branches. (**D**) Distal epithelial progenitor marker SOX9 expression becomes

*Figure 1 continued on next page*

*Figure 1 continued*

progressively restricted by day 6 of culture, correlating with a reduced number of proliferating KI67$^+$ cells. (**E**) Distal epithelial progenitor markers are upregulated by day 2 and downregulated by day 6. Expression values are normalized to transcript levels in isolated E13.5 distal epithelial tips. Each box depicts one biological replicate, showing the mean value among two technical replicates; 27 tip cultures were used for each time point. Scale bars: 100 μm (**A, C**), 50 μm (**D**). (**B**) Mean values are displayed; error bars represent S.D.

The online version of this article includes the following figure supplement(s) for figure 1:

**Source data 1.** Organoid forming efficiency counts, fresh vs. frozen tissue. Normalized expression values of progenitor markers.

**Figure supplement 1.** Growth factors, cellular composition and patterning of the organoids.

**Figure supplement 1—source data 1.** Normalized expression values of epithelial, mesenchymal, and endothelial markers.

---

levels on day 0, possibly reflecting the enrichment in distal characteristics or the endogenous activation of a distal developmental program between E13 and E14 (*Frank et al., 2019*). A marker of the proximal epithelium, *Sox2*, showed variable expression on day 0, suggesting that a variable number of proximal epithelial cells was isolated in the different experiments. *Sox2* was downregulated by day 2, suggesting that FGF10-supplemented media selectively support distal epithelial progenitors, as described previously (*Bellusci et al., 1997*; *Danopoulos et al., 2019*). Although SOX2 expression could be identified in the organoid core throughout the   six-day culture period, suggesting that proximal airway cells can be maintained in culture (*Figure 1—figure supplement 1C, D*), this region did not appear to expand compared to the distal SOX9 expression domain (*Figure 1—figure supplement 1C*), which may explain the progressive reduction in *Sox2* mRNA levels. Conversely, other markers of distal lung progenitors, such as *Id2*, *Bmp4*, and to a lesser extent *Sox9*, displayed upregulation by day 2, which may indicate expansion or conversion into distal progenitor cell identity. By day 6, all analyzed distal progenitor markers, as well as *Sox2*, were downregulated compared with E13.5 epithelial tips, suggesting that the distal progenitor cell identity is progressively lost during culture (*Figure 1D and E*). Since on day 6 epithelial branches appear to be patterned in the proximo-distal axis, are morphologically elongated, and have lost most proliferative capacity, similar to distal lung epithelium at late gestation (*Frank et al., 2019*), we hypothesized that distal epithelial progenitors undergo differentiation in culture.

## Collagen I promotes AT1 cell differentiation in culture

To assess epithelial differentiation, we performed whole-mount organoid immunostaining for SFTPC and RAGE, marking AT2 and AT1 cells, respectively. Low staining intensity was observed on days 0 and 1, indicating that within the first days of culture, the organoids are mainly composed of undifferentiated progenitors. On day 2, SFTPC/RAGE were co-expressed in a majority of the cells, similar to what is observed with alveolar bipotent progenitors (*Desai et al., 2014*; *Treutlein et al., 2014*). By day 6, two cell populations emerged: SFTPC$^-$/RAGE$^+$ cells in the stalk region of the elongated branches, and SFTPC$^+$/RAGE$^-$ cells in the tip region or at new sites of branching. The tip domain also comprised cells co-expressing SFTPC and RAGE, suggesting the maintenance of a bipotent progenitor (BP) population (*Treutlein et al., 2014*; *Figure 2A*). Quantification of the differentiated and progenitor cells on day 6 revealed approximately equal numbers of AT1 and BP cells in epithelial branches while AT2 cells were less prevalent (*Figure 2B*).

In the absence of Collagen I, the organoids displayed a reduced number of AT1 cells in favor of BP cells, indicating that a collagenous matrix (at 1:1 or 1:3 Matrigel/Collagen I ratio) promotes the differentiation of AT1 cells in vitro (*Figure 2B*), consistent with the reported role of the ECM in alveolar epithelial differentiation (*Kim et al., 2018*). We next asked to what extent the organoids represent mature distal lung tissue, and compared differentiation marker expression with distal lung tissue at E18.5, a time when differentiated alveolar epithelial cells can be unequivocally identified. The expression level of differentiated cell markers, including *Sftpc*, *Ager*, *Aqp5*, and *Hopx*, progressively increased until day 6, at which point they reached levels comparable with those observed in E18.5 distal lung tissue (*Figure 2C*). These data indicate that distal progenitors differentiate within six days of organoid culture, offering a simple model of early alveolar formation.

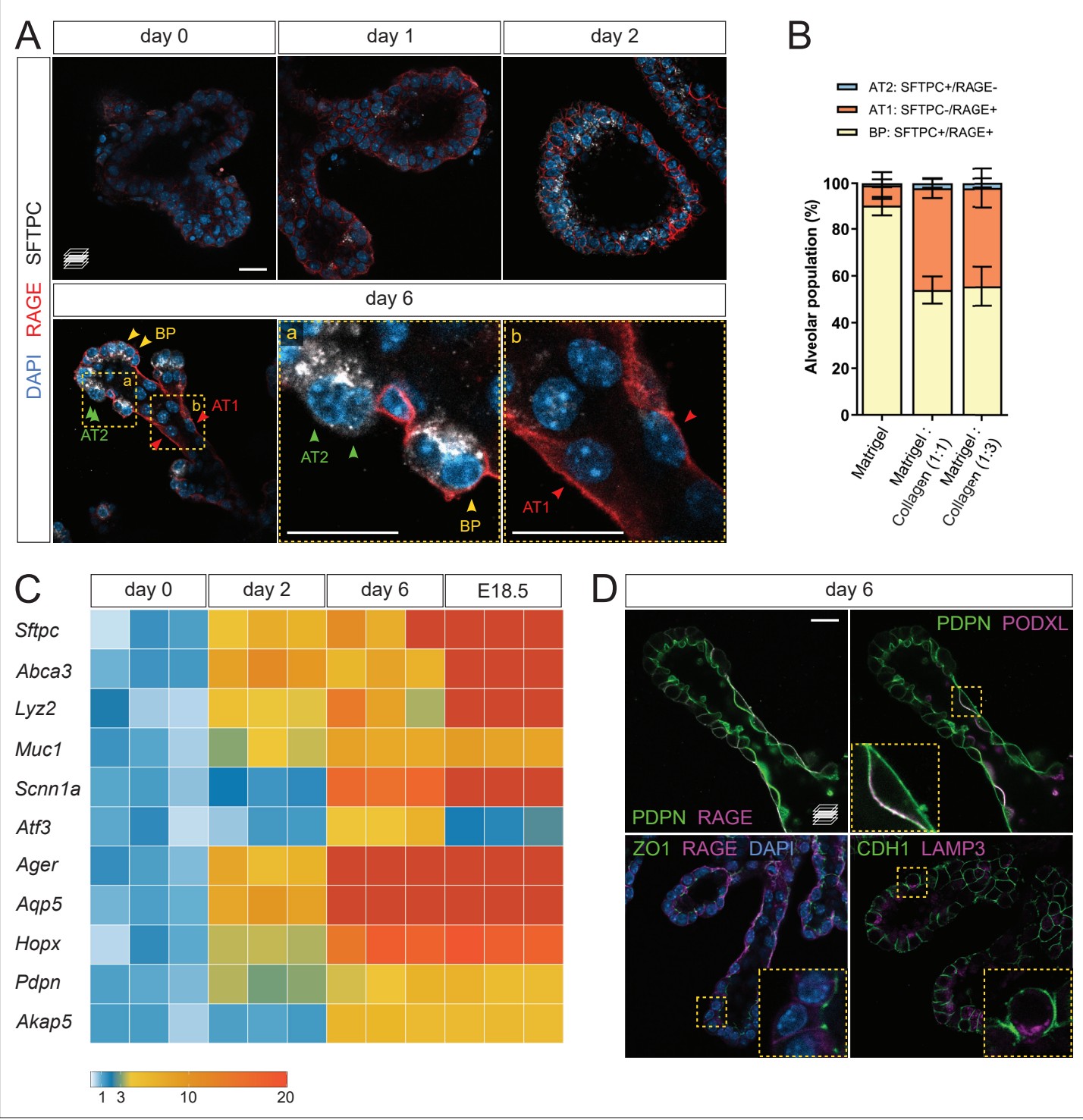

**Figure 2.** Lung distal progenitor cells differentiate in 3D explant cultures. (**A**) Whole-mount immunostaining for the alveolar cell markers SFTPC and RAGE at consecutive culture days. By day 2, SFTPC and RAGE are expressed by most cells. Day 6 epithelial branches comprise both bipotent progenitor (BP) cells (SFTPC+/RAGE+) and differentiated alveolar cells (AT1, SFTPC-/RAGE+ and AT2, SFTPC+/RAGE-) in an organotypic pattern. (**B**) Collagen I promotes AT1 cell differentiation, compared with Matrigel alone. Cell identities were scored on day 6. (**C**) Markers of differentiated AT1 and AT2 cells are transcriptionally upregulated by day 6 of culture, at levels comparable with E18.5 distal lung tissue. Expression values are normalized to culture day 0. Each box depicts one biological replicate, showing the mean value among two technical replicates; 27 tip cultures were used for each time point. (**D**) AT1 and AT2 cells in organoids are morphologically distinct and display apico-basal polarity. AT1 cells are elongated and often populate stalk regions. Upper row: AT1 cells display basolateral localization of the AT1 cell marker RAGE and apical localization of PODXL. Lower row: epithelial branches in

*Figure 2 continued on next page*

*Figure 2 continued*

organoids display adhesions typical of intact epithelia, such as tight junctions (ZO-1). The AT2 cell marker LAMP3 is localized to the apical secretory compartment of surfactant-producing cells. Scale bars: 20 µm. (**B**) Mean values are displayed; error bars represent S.D.

The online version of this article includes the following figure supplement(s) for figure 2:

**Source data 1.** Counts of alveolar cell types in organoid culture, Matrigel vs. Collagen. Normalized expression values of alveolar markers.

We further asked whether the AT1 and AT2 cells identified in the organoid cultures displayed morphological hallmarks of a differentiated respiratory epithelium. By immunostaining, we found that SFTPC⁻/RAGE⁺ cells displayed exclusive basolateral localization of RAGE (*Figure 2A and D*) and apical localization of Podocalyxin (PODXL) (*Figure 2D*), similar to observations in intact lung tissue at E19 (*Yang et al., 2016*). SFTPC⁻/RAGE⁺ cells were also marked by uniform plasma membrane expression of Podoplanin/T1alpha (PDPN) (*Figure 2D*). Immunostaining for Zonula Occludens-1 (ZO-1), a marker of tight junctions, labeled intercellular adhesions at the luminal/apical surface (*Figure 2D*) as previously reported (*Yang et al., 2016*), suggesting that the epithelial branches form an intact cellular monolayer and display barrier function. In addition, we found that Lysosome-Associated Membrane glycoProtein 3 (LAMP3), a lysosomal marker localized to lamellar bodies in differentiated AT2 cells (*Chang et al., 2013*; *Desai et al., 2014*), was localized in apical secretory organelles in SFTPC⁺ cells, indicating that these cells can differentiate into AT2 cells (*Figure 2D*). Altogether, these data show that cultured alveolar progenitors differentiate morphologically and molecularly into AT1 and AT2 cells, suggesting that fetal mouse alveolar organoids maintain the distal airway architecture found at late gestation stages (*Frank et al., 2019*).

## Casein Kinase modulates epithelial differentiation in lung organoids

We used fetal alveolar organoids to dissect the mechanisms by which WNT signaling controls the differentiation of AT1 and AT2 cells. To this end, we performed a chemical screen of WNT modulators from days 6 to 8 in the absence of supplemented growth factors (*Figure 3A*) and determined the relative alveolar cell composition by immunostaining for SFTPC and RAGE (*Figure 3B*). In control conditions, the proportion of SFTPC⁺/RAGE⁺ progenitors decreased from 54% (day 6, *Figure 2B*) to 40% (day 8, *Figure 3C*), with a concomitant increase in differentiated cell numbers (more pronounced for AT1 cells), suggesting that progenitors are actively differentiating in culture at the stages chosen for screening.

Unexpectedly, the majority of tested compounds (gray highlight) did not lead to significant changes in the proportion of AT1, AT2, or progenitor cells compared with DMSO controls (*Figure 3C*). Belonging to this group are first-in-class WNT pathway inhibitors such as IWR-1, IWP-2, and XAV-939 (*Chen et al., 2009*; *Huang et al., 2009*). Active WNT signaling is a feature of alveolar progenitors in homeostasis and regeneration, and blocking the WNT pathway by *Ctnnb1* knockout in AT2 cells was shown to induce AT1 differentiation at postnatal stages (*Frank et al., 2016*; *Nabhan et al., 2018*). In addition, an expansion of the AT1 cell population was observed in vitro after treatment of hiPSC-derived AT2 cells with XAV-939 (*Kanagaki et al., 2021*). Differentiation of cultured adult human alveolar progenitors (TM4SF1⁺) also showed WNT responsiveness, with the AT1 fate induced after XAV-939 treatment. In contrast with these data, the results of our chemical screen suggest that WNT inhibition via Axin stabilization is not sufficient to alter fate selection or cell differentiation of mouse fetal distal lung progenitors in our organoid cultures. This discrepancy with previous findings could be explained in part by differences in the phenotypic outcome of pharmacological versus genetic inhibition of WNT signaling, or by distinct regulation of the WNT pathway between pre- and postnatal stages. Limitations in our screening approach and the cellular composition of the organoid cultures could have influenced the results, as follows. First, a single concentration of 10 µM was used for all tested compounds, possibly leading to false negatives. Second, the heterogeneous cell composition of the organoids, in particular a variable number of mesenchymal cells, may have contributed to some of the variability in the data. Indeed, mesenchymal cells are fundamental players throughout lung development and engage in signaling crosstalk with epithelial cells, notably through FGF and WNT signaling (*McCulley et al., 2015*; *Volckaert and De Langhe, 2015*). Because of this variability in the relative cell composition of the organoids, four biological replicates were used for the screening experiments. Six compounds led to an increase in SFTPC⁺/RAGE⁺ BP cells (yellow highlight), including

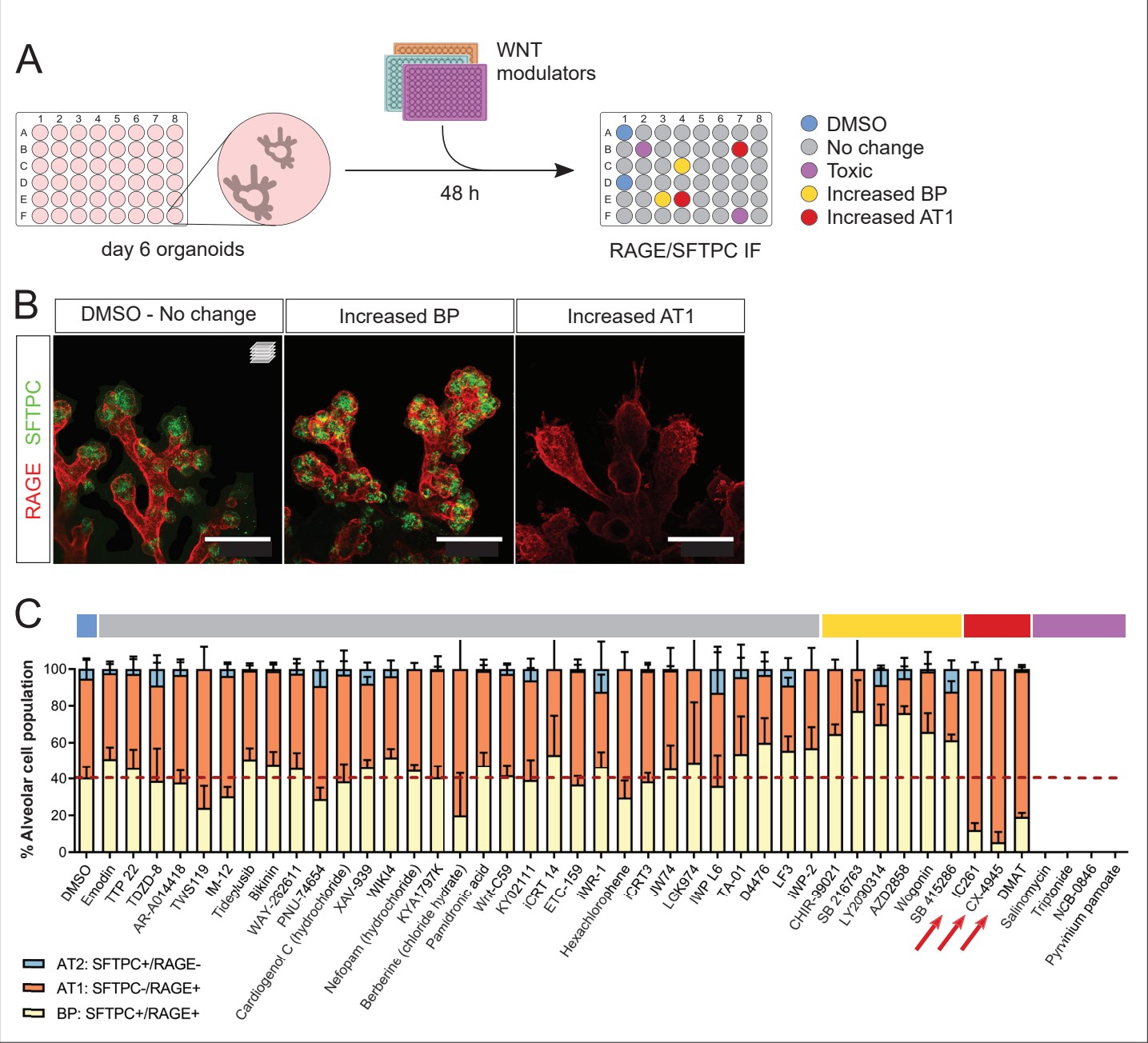

**Figure 3.** A WNT pathway chemical screen identifies modulators of alveolar cell differentiation. (**A**) Screening workflow schematic. Organoids at day 6 were treated for 48 h with WNT-modulating compounds. (10 μM). Phenotypes were scored by cell counting after immunostaining for SFTPC and RAGE and F-actin co-staining (Phalloidin). (**B**) Phenotypic classes observed in the chemical screen comprise an increased proportion of SFTPC$^+$/RAGE$^+$ cells (BP, middle panel) and increased proportion of SFTPC$^-$/RAGE$^+$ cells (AT1, right panel). (**C**) Casein Kinase inhibitors IC261, CX-4945, and DMAT (arrows) led to a higher proportion of AT1 cells at the expense of progenitors, while a majority of the compounds tested did not alter the proportion of differentiated alveolar cells in culture. Four biological replicates were analyzed, 12 tip cultures per compound. Scale bars: 50 μm. (**C**) Mean values are displayed; error bars represent S.D.

The online version of this article includes the following figure supplement(s) for figure 3:

**Source data 1.** Counts of alveolar cell types in the WNT modulators screen. Percentages of alveolar cell types in the WNT modulators screen.

several Glycogen Synthase Kinase 3 (GSK3) inhibitors such as CHIR-99021, and a natural flavonoid, Wogonin. In line with published data (*Ostrin et al., 2018*; *de Carvalho et al., 2019*), we found that sustained WNT activity was associated with progenitor maintenance. Finally, three compounds induced a greater percentage of AT1 cells (SFTPC$^-$/RAGE$^+$) at the expense of bipotent progenitors

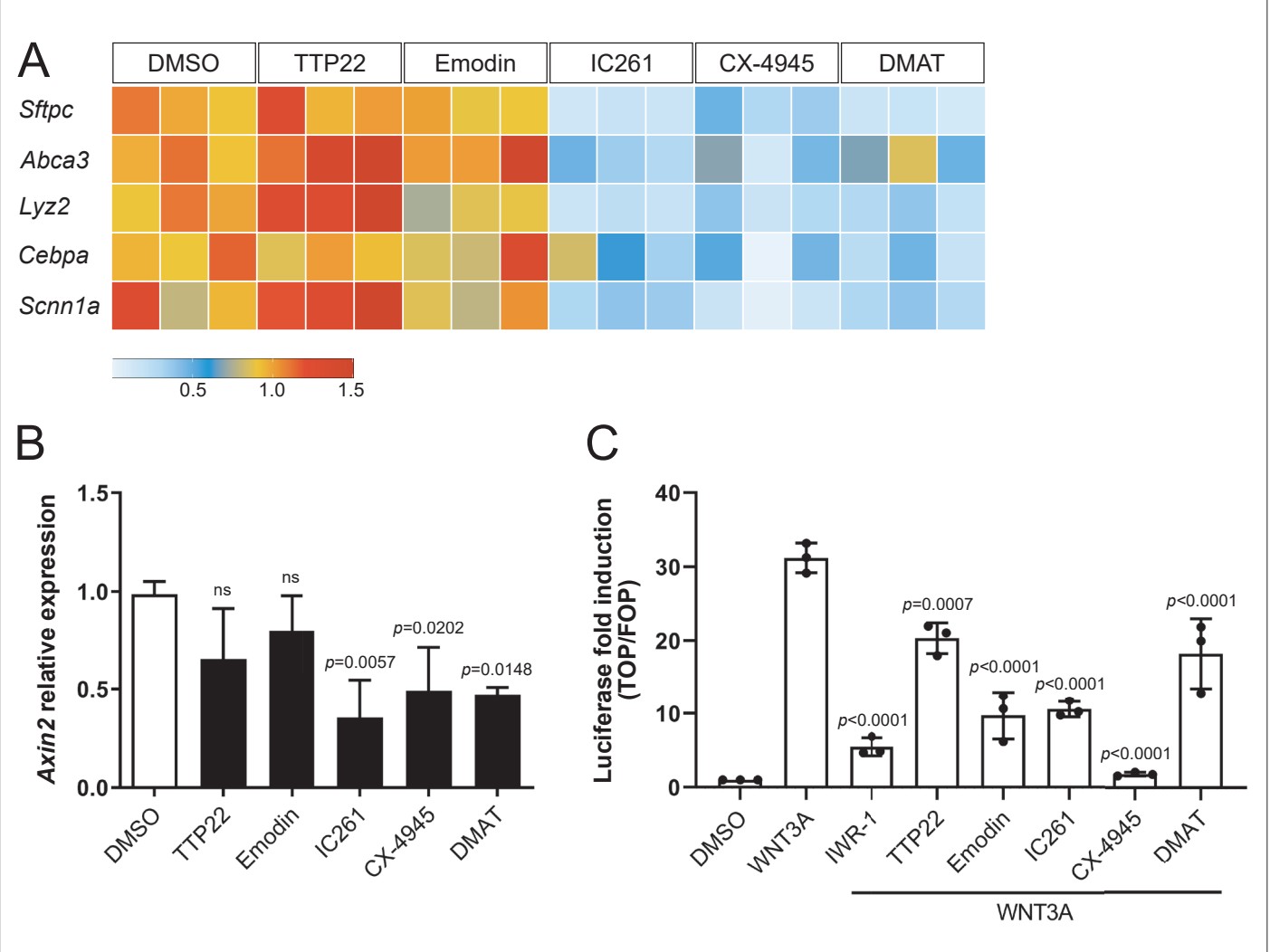

**Figure 4.** Compounds promoting AT1 cell differentiation lead to concomitant loss of AT2 marker gene expression and WNT-dependent transcription. (**A**) IC261, CX-4945, and DMAT treatments (48 h) lead to downregulation of AT2 cell markers. TTP22 and Emodin (also CK2 inhibitors) do not lead to the same phenotype, possibly due to differences in potency/target affinity. Expression levels relative to DMSO control. Three biological replicates (consisting of 2 technical replicates each) were analyzed per compound. (**B**) Casein Kinase inhibition by IC261, CX-4945, and DMAT (6 h) leads to reduced *Axin2* relative expression. Treatment with TTP22 or Emodin does not significantly affect *Axin2* levels. More than three biological replicates were analyzed. (**C**) CK inhibition reduced WNT/$\beta$-catenin-dependent transcription in HEK293T epithelial cells (SuperTOPFlash-based luciferase assay). (**B**) Mean values are displayed; error bars represent S.D.; p-values from one-way ANOVA, Tukey's multiple comparisons test. (**C**) Mean values displayed; error bars represent S.D.; p-values from one-way ANOVA, Tukey's multiple comparisons test; displayed p-values refer to comparisons with WNT3A-treated condition.

The online version of this article includes the following figure supplement(s) for figure 4:

**Source data 1.** Normalized expression values of AT2 cell markers, CK inhibitors vs. DMSO control. Normalized *Axin2* expression values, CK inhibitors vs. DMSO control and related statistics. Raw luciferase data. Normalized luciferase values and related statistics.

(*Figure 3C*). Mechanistically, these compounds inhibit Casein Kinase (CK) 1 (IC261) and CK2 (CX-4945, DMAT), central regulators of WNT signaling and cell cycle progression (*St-Denis and Litchfield, 2009*; *Cruciat, 2014*; *Venerando et al., 2014*).

## Casein Kinase inhibition leads to downregulation of AT2 marker gene expression and WNT-dependent transcription

Casein Kinase (CK) acts in stem cell maintenance by phosphorylating transcriptional regulators, WNT signaling components, and cell cycle factors (*St-Denis and Litchfield, 2009*; *Cruciat, 2014*). We

found that CK inhibition by IC261, CX-4945, and DMAT led to the downregulation of markers of the AT2 fate, including *Sftpc* and *Abca3*, compared with control organoids (***Figure 4A***), suggesting that upon CK inhibition, progenitor cells lose AT2 fate potential and convert to AT1 cells. Similarly, WNT signaling activation was reduced in organoids treated with the same CK inhibitors for 6 h as assessed by measuring mRNA levels of *Axin2* (***Figure 4B***), a β-catenin-dependent WNT target gene. Pharmacological CK inhibition could also significantly reduce the transcriptional response to WNT/β-catenin signaling in HEK293T cells, as shown by SuperTOPFlash luciferase reporter assays (***Figure 4C***). Collectively, these data indicate that progenitor cells convert to AT1 cells by combined loss of β-catenin-dependent WNT activity and AT2 marker expression, and suggest that Casein Kinases antagonize cell differentiation by modulating WNT signaling transduction and transcriptional regulation. However, since Casein Kinases mediate a plethora of housekeeping cellular functions, determining which specific CK targets are involved in alveolar differentiation requires further investigation. Recently, CK2 has emerged as a prominent target of SARS-CoV-2 infection in the lung, and its overactivation was associated with increased viral transmission by cytoskeletal remodeling and formation of cell protrusions mediating virus budding (***Bouhaddou et al., 2020***). Combined with our data, these recent findings suggest that SARS-CoV-2 infection could affect lung cell fate establishment and regeneration of the alveolar compartment directly by targeting CK2 activity.

In summary, our fetal mouse lung alveolar organoids constitute an ideal system to assess cell differentiation at the single-cell level and will allow the investigation of this process by live imaging and chemical screening. This new model should also facilitate the identification of modulators of alveolar cell differentiation, and help discover treatments to promote lung development and maturation in premature newborns.

# Materials and methods

**Key resources table**

| Reagent type (species) or resource | Designation | Source or reference | Identifiers | Additional information |
|---|---|---|---|---|
| Strain, strain background (*Mus musculus*) | C57BL/6 J mice | https://www.jax.org/strain/000664 | RRID:IMSR_JAX:000664 | |
| Cell line (*Homo sapiens*) | HEK-293T | ATCC | ATCC:CRL-3216; RRID:CVCL_0063 | |
| Other | Rat-tail Collagen I | Corning | Corning:354236 | |
| Other | Growth factor reduced Matrigel | Corning | Corning:356231 | |
| Other | DMEM/F12 | Sigma | Sigma:D6434 | |
| Other | L-Glutamine | Sigma | Sigma:G7513 | |
| Other | Penicillin/Streptomycin | Sigma | Sigma:P4333 | |
| Other | DNAse I | Roche | Roche:10104159001 | |
| Other | 10% BSA | Sigma | Sigma:A1595 | |
| Other | MEM Non-Essential Amino Acids solution | Gibco | Thermo Fisher Scientific:11140050 | |
| Other | Insulin-Transferrin-Selenium-Ethanolamine (ITS-X) | Gibco | Thermo Fisher Scientific:51500056 | |
| Other | Primocin | Invivogen | Invivogen:ant-pm-1 | |
| Other | 10 x DMEM | Sigma | Sigma:D2429 | |
| Chemical compound, drug | 1 N NaOH | Sigma | Sigma:S2770 | |
| Peptide, recombinant protein | Human recombinant FGF10 | R&D Systems | R&D Systems:345-FG | |
| Peptide, recombinant protein | Human recombinant FGF7 | Peprotech | Peprotech:100–19 | |

*Continued on next page*

*Continued*

| Reagent type (species) or resource | Designation | Source or reference | Identifiers | Additional information |
|---|---|---|---|---|
| Other | CryoStor CS10 | Stem Cell Technologies | Stem Cell Technologies:07959 | |
| Other | Normal donkey serum | Jackson ImmunoResearch | Jackson Immuno Research:017-000-121; RRID:AB_2337258 | |
| Antibody | Anti-SOX9 (rabbit polyclonal) | Millipore | Millipore:AB5535; RRID:AB_2239761 | (1:250) |
| Antibody | Anti-KI67 (rat monoclonal) | Invitrogen | Thermo Fisher Scientific:14-5698-82; RRID:AB_10854564 | (1:250) |
| Antibody | Anti-SOX2 (goat polyclonal) | R&D Systems | R&D Systems:AF2018; RRID:AB_355110 | (1:250) |
| Antibody | Anti-KRT5 (rabbit polyclonal) | Abcam | Abcam:ab53121; RRID:AB_869889 | (1:250) |
| Antibody | Anti-FOXJ1 (mouse monoclonal) | Invitrogen | Thermo Fisher Scientific:14-9965-82; RRID:AB_1548835 | (1:250) |
| Antibody | Anti-ProSP-C (rabbit polyclonal) | Millipore | Millipore:AB3786; RRID:AB_91588 | (1:500) |
| Antibody | Anti-RAGE (rat monoclonal) | R&D Systems | R&D Systems:MAB1179; RRID:AB_2289349 | (1:250) |
| Antibody | Anti-PDPN (sheep polyclonal) | R&D Systems | R&D Systems:AF3670; RRID:AB_2162070 | (1:250) |
| Antibody | Anti-PDPN (syrian hamster monoclonal) | DSHB | DSHB:8.1.1; RRID:AB_531893 | (1:20) |
| Antibody | Anti-PODXL (goat polyclonal) | R&D System | R&D Systems:AF1556; RRID:AB_354858 | (1:250) |
| Antibody | Anti-ZO1/TJP1 (mouse monoclonal) | Invitrogen | Thermo Fisher Scientific:33–9100; RRID:AB_2533147 | (1:250) |
| Antibody | Anti-CDH1 (goat polyclonal) | R&D Systems | R&D Systems:AF748; RRID:AB_355568 | (1:250) |
| Antibody | Anti-LAMP3 (rat monoclonal) | Dendritics | IMGENEX:DDX0192; RRID:AB_1148779 | (1:250) |
| other | Phalloidin, Alexa Fluor 488 conjugate | Invitrogen Thermo Fisher Scientific:A12379 | Thermo Fisher Scientific:A12379 | (1:500) |
| other | DAPI | Sigma | Sigma:D9542 | (1 µg/ml) |
| Antibody | Alexa 488-, 568-, 647- or Cy3-conjugated secondaries | Invitrogen | | (1:500) |
| Antibody | Alexa 488-, 568-, 647- or Cy3-conjugated secondaries | Jackson ImmunoResearch | | (1:500) |
| Other | DMEM + Glutamax | Gibco | Thermo Fisher Scientific:31966021 | |
| Other | FBS superior | Sigma | Sigma:S0615 | |
| Other | Lipofectamine 3000 Transfection Reagent | Invitrogen | Thermo Fisher Scientific:L3000001 | |
| Transfected construct (*Homo sapiens*) | M50 Super 8 x TOPFlash (plasmid) | Addgene | Addgene:12456; http://n2t.net/addgene:12456; RRID:Addgene_12456 | |
| Transfected construct (*Homo sapiens*) | M51 Super 8 x FOPFlash (TOPFlash mutant) (plasmid) | Addgene | Addgene:12457; http://n2t.net/addgene:12457; RRID:Addgene_12457 | |
| Transfected construct (*Homo sapiens*) | pRL-TK (plasmid) | Promega | Promega:E2241 | |
| Peptide, recombinant protein | Recombinant human WNT3A | Proteintech | Proteintech:HZ-1296 | (500 ng/ml) |
| Chemical compound, drug | Dimethyl sulfoxide (DMSO) | Sigma | Sigma:D2650 | |

*Continued*

| Reagent type (species) or resource | Designation | Source or reference | Identifiers | Additional information |
|---|---|---|---|---|
| Chemical compound, drug | Emodin | Tocris | Tocris:3811 | (10 µM) |
| Chemical compound, drug | TTP22 | Tocris | Tocris:4432 | (10 µM) |
| Chemical compound, drug | IC261 | Sigma | Sigma:I0658 | (10 µM) |
| Chemical compound, drug | CX-4945 | Enzo Life Sciences | Enzo Life Sciences: ENZ-CHM151 | (10 µM) |
| Chemical compound, drug | DMAT | Sigma | Sigma:SML2044 | (10 µM) |
| Chemical compound, drug | IWR-1 | Sigma | Sigma:I0161 | (10 µM) |
| Commercial assay or kit | Dual-Luciferase Reporter Assay System | Promega | Promega:E1910 | |
| Commercial assay or kit | NucleoSpin RNA kit | Macherey-Nagel | Macherey-Nagel:740955.50 | |
| Commercial assay or kit | Superscript III Reverse Transcriptase system | Invitrogen | Thermo Fisher Scientific:18080093 | |
| Commercial assay or kit | DyNAmo ColorFlash SYBR green qPCR kit | Thermo Scientific | Thermo Fisher Scientific:F416XL | |
| Sequence-based reagent | qPCR | This paper | *Supplementary file 1* | |
| Software, algorithm | Fiji/ImageJ 1.53 c | *Schindelin et al., 2012*; doi:10.1038/nmeth.2019 | RRID:SCR_002285 | |
| Software, algorithm | GraphPad Prism 8 | GraphPad | RRID:SCR_002798 | |

## Media preparation

Frozen aliquots of rat-tail Collagen I (500 µl, Corning 354236) and growth factor reduced (GFR) Matrigel (550 µl, Corning 356231) were thawed on ice during lung epithelial tip isolation. One aliquot of Collagen I and Matrigel were sufficient for two dissociations (two 48-well plates).

Dissection Medium (DM) composition: DMEM/F12 (Sigma D6434), 1 mM L-Glutamine (Sigma G7513), 1 x Penicillin/Streptomycin (Sigma P4333), 100 µg/ml DNAse I (Roche 10104159001).

Complete Medium (CM) composition: DMEM/F12, 1 mM L-Glutamine, 0.25% BSA (Sigma A1595), 1 x MEM Non Essential Amino Acids (Gibco 11140050), 0.1 x Insulin-Transferrin-Selenium-Ethanolamine (Gibco 51500056), 100 µg/ml Primocin (Invivogen ant-pm-1).

Media were prepared fresh just before tissue isolation.

## Lung epithelial tip isolation

E14.5 C57BL/6 J mouse embryos (staged according to *Theiler, 1989*) were harvested and washed in ice-cold PBS. Embryos were quickly decapitated and lungs extracted and washed in fresh ice-cold PBS. Lobes were isolated by cutting them away from the proximal bronchi using microscissors (Fine Science Tools 15003–08).

Using a P1000 pipette and tips coated with fresh 2% BSA (Sigma A1595), lobes from three to five lungs were transferred to freshly prepared ice-cold DM in a 60 mm petri dish. Lobes were pipetted up and down using a coated-tip P1000 pipette until fragmented. Next, if necessary, a coated-tip P200 pipette was used to triturate the tissue more finely. The degree of dissociation was monitored using a Zeiss Stemi 305 stereomicroscope. The petri dish was placed over ice-cold black metal blocks, allowing for simultaneous temperature control and contrast for visual inspection.

Fragments including one to threeepithelial tips were collected using a coated-tip P20 pipette and transferred in fresh DM on ice, to dilute out dissociated mesenchymal cells. Epithelial tips were re-collected, counted and transferred into a 1.5 ml tube on ice (about 100–120 per isolation). Tips were allowed to sink to the bottom of the tube on ice for 1–2 min and then washed twice in 100 µl DM and twice in 100 µl CM.

## Organoid embedding
During epithelial tips washes, thawed Collagen I (500 µl) was neutralized by adding 57 µl 10 x DMEM (Sigma D2429) and approximately 16–17 µl 1 M NaOH (Sigma S2770). Once mixed to homogeneity, neutralized collagen was allowed to polymerize on ice for 10 min. Next, neutralized collagen was mixed to an isovolume of GFR Matrigel and kept on ice.

Washed epithelial tips were resuspended in 500 µl ice-cold CM, to which 500 µl of Matrigel/Collagen solution was added. Epithelial tips were carefully and thoroughly resuspended on ice and plated using cooled P20 tips onto a 48-well tissue culture plate (20 µl/well) over a 39 °C heat block. Domes were allowed to solidify over the heat block for 30 min and plates were moved to a standard tissue culture incubator at 37 °C for 45 min-1 h before CM with growth factors (CM+) was added as follows.

## Organoid culture
A total of 50 ml CM was warmed up to 37 °C and human recombinant FGF proteins were added (50 ng/ml FGF10, R&D 345-FG and 25 ng/ml FGF7, Peprotech 100–19). 300 µl CM with growth factors (CM+) was added to each culture well.

At day 3, organoid medium was changed to diluted CM+ (1:1 with CM, halved growth factor concentration).

At day 6, organoid medium was changed to CM without supplemented growth factors.

For long-term maintenance of organoid cultures, organoid medium was changed every two days starting at day 6. Organoids could be maintained in media devoid of growth factors for up to 30 days.

Organoids with overgrown mesenchymal cells (attached to the plastic) were excluded from further analysis.

## RT-qPCR
Total RNA was isolated from pooled organoids (9–15 domes) or tissue using the NucleoSpin RNA kit (Macherey-Nagel 740955.50). cDNA was synthesized from total RNA using the Superscript III Reverse Transcriptase system (Thermo Scientific 18080093). qPCR was performed using the DyNAmo Color-Flash SYBR green qPCR kit (Thermo Scientific F416XL) on a CFX Connect Real-Time System (Bio-Rad). qPCR reactions were performed in technical duplicates; data from at least three biological replicates were collected. Gene expression values were normalized to the mouse *Actb* gene. Heatmaps were generated using R/Bioconductor and ggplot2 packages. Mean values among technical replicates were color-coded and each biological replicate was represented.

## Organoid whole-mount immunostaining
Matrix domes were quickly washed in PBS and fixed in 4% PFA for 15 min at room temperature (RT). Domes were detached by the well bottom with a flat microspatula and transferred to tubes or staining baskets. After several PBS washes, domes were permeabilized for 1 h at RT in sterile-filtered permeabilization solution: 0.5% Triton-X100, 0.5% Tween20, 3% Donkey Serum (Jackson ImmunoResearch 017-000-121), 1% BSA. Samples were incubated with primary antibodies in permeabilization solution diluted 1:1 in PBS for two days at 4 °C, with mild agitation. Domes were then washed in PBST (PBS + 0.05% Tween20) for 6 h at RT or overnight at 4 °C with mild agitation. Next, domes were incubated with secondary antibodies, Alexa488-Phalloidin (Invitrogen A12379, both at 1:500) and DAPI (1 µg/ml) in diluted permeabilization solution (1:1 in PBS) for 3 h at RT, followed by 3–4 h wash in PBST. Domes were mounted in mounting medium (Dako S3023) on glass slides using a thin vacuum grease ring as coverslip spacer. After overnight incubation at 4 °C, imaging was performed on a Leica SP8 or Zeiss LSM 800 Observer confocal microscope, ×25 and ×40 magnification.

## Antibodies
The following primary antibodies were used: rabbit anti-SOX9 (Millipore AB5535), rabbit anti-ProSP-C (Millipore AB3786), rat anti-RAGE (R&D MAB1179), goat anti-PDPN (R&D AF3670), syrian hamster

anti-PDPN (DSHB 8.1.1-SN), goat anti-PODXL (R&D AF1556), mouse anti ZO-1/TJP1 (Invitrogen 33–9100), goat anti-CDH1 (R&D AF748), rat anti-LAMP3 (Dendritics DDX0192), rat anti-KI67 (Invitrogen 14-5698-82), goat anti-SOX2 (R&D AF2018), mouse anti-FOXJ1 (Invitrogen 14-9965-82), rabbit anti-KRT5 (Abcam ab53121). All primary antibodies were used at 1:250 dilution, except anti-ProSP-C (1:500). All secondary antibodies (Invitrogen, Jackson ImmunoResearch) were used at 1:500 dilution.

### Cryostorage
Freshly isolated E14.5 epithelial tips were mixed with 500 µl of CryoStor CS10 (Stem Cell Technologies 07959) and stored at –80 °C for up to 6 months.

### Chemical screen
Organoids were plated into 48-well plates and incubated from day 6 for 48 h in the presence of compounds. Chemicals were dissolved in CM devoid of additional growth factors at a final concentration of 10 µM (from a DMSO stock). Screened compounds belong to the Stem Cell Signaling Compound Library (MCE HY-L017). Treatment with 0.1% DMSO (Sigma D2650) was used as internal control in each organoid plate. On day 8, organoids were washed and fixed in 4% PFA and three or four technical replicates were pooled for immunostaining. Confocal scans were followed by cell counting as detailed below. Four biological replicates (12 tip cultures in total) were analyzed per compound.

### Cell counts
For each confocal stack, three optical sections were selected corresponding to 25, 50, and 75% of the confocal Z stack range. For each of these planes, SFTPC$^+$/RAGE$^+$, SFTPC$^-$/RAGE$^+$, and SFTPC$^+$/RAGE$^-$ cells were manually counted using the Cell Counter plugin in Fiji/ImageJ (*Schindelin et al., 2012*) and quantified as a percentage of DAPI$^+$ cells. Percentages were averaged among the three planes, and mean alveolar cell percentages of at least nine organoids among three technical replicates derived from at least three biological replicates (different pregnant dams) were then plotted and error was calculated as standard deviation. Day 6 tip cultures contained on average 54% SFTPC$^+$/RAGE$^+$ cells, 43.9% SFTPC$^-$/RAGE$^+$ cells, 2.1% SFTPC$^+$/RAGE$^-$ cells (*Figure 2—source data 1*). Tip cultures treated from day 6 to day 8 with 0.1% DMSO contained on average 40.5% SFTPC$^+$/RAGE$^+$ cells, 54.1% SFTPC$^-$/RAGE$^+$ cells, 5.4% SFTPC$^+$/RAGE$^-$ cells (*Figure 3—source data 1*).

### Cell lines
Human Embryonic Kidney cells (HEK293T, ATCC CRL-3216) were certified by STR profiling by ATCC and tested negative for mycoplasma contamination. Cells were maintained in DMEM (Gibco 31966021), 10% FBS (Sigma S0615), 1% Penicillin/Streptomycin and passaged using TrypLE Express (Gibco 12604021).

### Luciferase assay
SuperTOPFlash reporter assays were performed as in *Veeman et al., 2003*. In brief, 15,000 HEK293T cells/well were seeded into gelatin-coated 96-well plates and allowed to grow overnight. Cells were transfected with a combination of 100 ng SuperTOPFlash or 100 ng SuperFOPFlash and 10 ng pRL-TK (Promega E2241) plasmids. M50 Super 8 x TOPFlash and M51 Super 8 x FOPFlash (TOPFlash mutant) were gifts from Randall Moon (Addgene plasmid # 12456; http://n2t.net/addgene:12456; RRID:Addgene_12456 and Addgene plasmid # 12457; http://n2t.net/addgene:12457; RRID:Addgene_12457). Transfections were carried out using Lipofectamine 3000 Reagent (Invitrogen L3000001) according to manufacturer's instructions. Transfected cells were serum-starved for 6 h and treated with 500 ng/ml human recombinant WNT3A (Proteintech HZ-1296) and 10 µM chemical compounds or 0.1% DMSO for 12 h. Cells were washed with PBS on ice to remove phenol red and lysed. Firefly and Renilla luciferase assays were carried out using Dual-Luciferase Reporter Assay System (Promega E1910) in technical triplicates. Raw Firefly luminescence values were normalized to corresponding Renilla values. Technical triplicates were averaged, normalized over FOPFlash values and normalized over DMSO-treated control (no WNT3A stimulation). Experiments were carried out in biological triplicates.

## Acknowledgements

We thank Emma Rawlins, Saverio Bellusci and Chi-Chung Wu for discussions, and Saverio Bellusci, Chi-Chung Wu, Felix Gunawan and Simon Perathoner for suggestions and comments on the manuscript. We thank Hyun-Taek Kim, Alessandra Gentile and Till Lautenschläger for technical suggestions and help. Paolo Panza and Didier Stainier are recipients of a CPI flexible outbreak project grant. Research in the Stainier lab is supported in part by the Max Planck Society, the DFG (Sonderforschungsbereich) (SFB 834), the Leducq Foundation, and the European Research Council (AdG project: ZMOD 694455).

## Additional information

### Competing interests

Didier YR Stainier: Senior editor, eLife. The other authors declare that no competing interests exist.

### Funding

| Funder | Grant reference number | Author |
| --- | --- | --- |
| Max-Planck-Gesellschaft | | Konstantinos Gkatzis<br>Paolo Panza<br>Sofia Peruzzo<br>Didier YR Stainier |
| Cardio-Pulmonary Institute | Flexible outbreak project grant | Paolo Panza<br>Didier YR Stainier |
| Deutsche Forschungsgemeinschaft | SFB 834 | Didier YR Stainier |
| Leducq Foundation | | Didier YR Stainier |
| H2020 European Research Council | ZMOD 694455 | Didier YR Stainier |

The funders had no role in study design, data collection and interpretation, or the decision to submit the work for publication.

### Author contributions

Konstantinos Gkatzis, Conceptualization, Formal analysis, Investigation, Methodology, Validation, Visualization, Writing – original draft, Writing – review and editing; Paolo Panza, Conceptualization, Formal analysis, Funding acquisition, Investigation, Methodology, Validation, Visualization, Writing – original draft, Writing – review and editing; Sofia Peruzzo, Formal analysis, Investigation, Validation, Writing – review and editing; Didier YR Stainier, Conceptualization, Funding acquisition, Supervision, Writing – original draft, Writing – review and editing

### Author ORCIDs

Konstantinos Gkatzis http://orcid.org/0000-0003-2152-0817
Paolo Panza http://orcid.org/0000-0002-8702-5504
Sofia Peruzzo http://orcid.org/0000-0003-0008-6361
Didier YR Stainier http://orcid.org/0000-0002-0382-0026

### Decision letter and Author response

Decision letter https://doi.org/10.7554/eLife.65811.sa1
Author response https://doi.org/10.7554/eLife.65811.sa2

## Additional files

### Supplementary files

- Supplementary file 1. Table of qPCR primers.
- Transparent reporting form

## Data availability

All data generated or analyzed during this study are included in the manuscript and supporting files. Source data files have been provided for Figures 1, 2, 3, 4 and Figure 1-figure supplement 1.

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
