## [Decision Letter]

**Acceptance summary:**

This manuscript describes a newly developed method to culture mouse lung distal progenitor cells in a 3D serum-free condition. By modulating the concentration of supplied growth factors, distal lung progenitor cells could branch and then gradually differentiate into AT1 and AT2 cells, a process very similar to alveolar development in mice. The authors also found that collagen I could significantly promote progenitor cells to differentiate into AT1 cells. Based on this culture system, the authors performed a Wnt signaling-related-drug screening and identified that casein kinases are essential for the specification of alveolar epithelial cells. The manuscript is well-written and discusses the method and its potential limitations nicely. This is an important study and will be of interest to the readership of *eLife*.

**Decision letter after peer review:**

Thank you for submitting your article "Differentiation of mouse fetal lung alveolar progenitors in serum-free organotypic cultures" for consideration by *eLife*. Your article has been reviewed by three peer reviewers, and the evaluation has been overseen by a Reviewing Editor and Edward Morrisey as the Senior Editor. The following individuals involved in review of your submission have agreed to reveal their identity: Aimee K Ryan (Reviewer #2); Amy Firth (Reviewer #3).

The reviewers have raised several important questions about the model system, which should be addressed and integrated into the revised discussion. Please address specifically the questions raised with respect to the potential role of mesenchymal/other non-epithelial cells in the organoid system, the effect of Wnt inhibitors on cell fate specification, and provide additional data to further strengthen the Wnt readouts upon screening and the CK/Wnt link.

Essential revisions:

(1) Additional experimental data on Wnt readouts and Wnt/CK link

(2) Discussion on the potential role of mesenchymal/other non-epithelial cells in the organoid system

(3) Discussion on the effect of Wnt inhibitors on cell fate specification

(4) Data/discussion on potential *Sox2* differentiation

*Reviewer #1:*

In this manuscript, the authors presented a newly developed method that the mouse lung distal progenitor cells can be cultured in a 3D serum-free condition. By modulating the concentration of supplied growth factors, these distal progenitor cells could branch and then gradually differentiate into AT1 and AT2 cells, a process is very similar to the alveolar development process in mice. The authors also found that collagen I could significantly promoted progenitor cells to differentiate into AT1 cells. Based on this culture system, the authors performed a Wnt signaling-related-drug screening and identified that casein kinases are essential for the specification of alveolar epithelial cells. Modeling alveolar development *in vitro* is very important to facilitate future studies on the cellular and molecular mechanisms of alveolar development.

1. In this culture system, the authors found that distal progenitor cells could expand and become alveolar lineage epithelial cells (bipotent cells, AT1 and AT2). Considering that distal progenitor cells at E14.5 also could also differentiate into proximal airway lineage cells, the authors should provide information on whether these cells could differentiate into *Sox2*^+^ proximal progenitor cells and other proximal cell types.

2. Based on their dissection method, the distal progenitor cells-derived organoids may contain mesenchymal cells or other non-epithelial cell types, which may significantly influence the proliferation and differentiation of distal progenitor epithelial cells. The authors should at least consider this possibility and discuss whether these drug-induced phenotypes are caused by modulation of mesenchymal cells or other non-epithelial cell types.

3. The electron microscopy analysis can better demonstrate the cell identity of bipotent cells, AT1 and AT2 cells in the organoids.

4. Using this culture system to perform a Wnt signaling-related drug screening experiment, the authors found that Wnt inhibitors do not affect the cell fate specification, which does not agree with previous results. The authors should provide a discussion. Further details of this drug screening experiment should be provided in the manuscript.

5. The link between Casein Kinase (CK) inhibition and Axin2 mRNA expression is not strong enough. The authors should provide more evidence to support their observed relationship between CK and Wnt signaling. Furthermore, the authors should discuss and propose the mechanisms of CK in regulating alveolar epithelial cell fate specification.

*Reviewer #2:*

The authors have developed an ex vivo organoid culture system using distal epithelial fragments from the mouse lung that permits investigation of studied cell type differentiation and that can be manipulated to evaluate the role of signaling pathways on the process of alveolization. The organoid cultures were established using manually dissected and dissociated distal epithelial fragments from E14.5 mouse lungs. At this point of development, pseudoglandular development is almost complete and alveolar progenitors have been specified. In this manuscript the authors have defined culture conditions under which the tissue fragments can develop in three-dimensional culture to form an epithelial structure with a lumen. Within 6 days of culture the organoids formed from the distal lung epithelial fragments are capable of forming epithelial branches and undergo cell type differentiation similar to what is observed *in vivo*. The observed epithelial growth and lumenization occurs in the absence of hyperproliferation of mesenchymal cells. Of interest is the observation that tissue that has been cryopreserved can also be expanded to form organoids, albeit with a reduced efficiency.

During the six day culture period, the organoids assume lung-like structures and display epithelial buds and branches. Sufficient details are provided to permit others to establish the lung organoids in their own labs. Morphological analysis of these structures demonstrates that the distal fragments are able to reorganize to form a closed lumen and expand the number of epithelial branches within an individual organoid. Throughout the culture period, the differentiation status of cells within the organoids were assayed using immunofluorescence to identify SOX9 +ve progenitor cells and differentiated alveolar Type 1 and Type 2 cells and quantitative RT-PCR to measure changes in expression levels of distal epithelial markers. Expression of SOX9 and ID2 were analyzed to identify the distal tip progenitor population. SOX9 +ve progenitors were initially observed at the tips on day 0, became widespread by day 1 and localized to discrete patches on day 2 before becoming once again restricted to the distal tips of the epithelial branches on day 6. The increased number of cells expressing SOX9 is supported by the increase in *Sox9* mRNA expression observed at day 2. It is not clear if the changes in expression and localization of SOX9 positive cells reflect the reorganization of cells when the distal tips are place in organoid culture, a general transient increase in cells assuming a progenitor cell phenotype or if they reflect differences in the proportion of progenitor cells in the tissue used for the organoid culture versus the E13.5 lung distal tips.

Gene expression analysis of the differentiation of alveolar cell types AT1 and AT2 in the distal epithelium organoids and show similar gene expression levels as compared to E18.5 embryonic lungs. The cells in the organoid epithelium express AT1 and AT2-specific marker proteins and are polarized with apical localization of tight and adherens junction proteins. The presence of collagen clearly promotes the differentiation of bipotential progenitors to differentiate into AT1 cells. There also appears to be an increase in AT2 positive cells but these are a much smaller proportion of the total cell population.

The effect of altering WNT signaling on cell type differentiation is determined using the organoid cultures in a chemical screen. Chemicals are categorized based on their ability to shift the proportion of bipotential progenitors versus differentiated AT1 cells or their toxicity. The effects on cell type differentiation were confirmed by quantitative RT-PCR. The principal findings from this study is that sustaining WNT activity promoted maintenance of the progenitor population. In contrast, compounds that inhibit Casein Kinase 1 or 2, central regulators of WNT signaling and cell cycle progression pushed differentiation of the progenitor population in the direction of AT1 cells. This was associated with an inhibition of expression of genes associated with AT2 cell fate and decreased WNT signaling, as assessed by decreased mRNA expression of the Β-catenin target gene Axin2.

In summary, this manuscript describes the development of an organoid culture method for distal lung epithelium and demonstrated its applicability for understanding the role of ECM and WNT signaling in differentiated cell types. The branching process was not examined in detail but this culture system should permit further analysis of the molecular and morphological events that underlie this process and to compare to what has been determined *in vivo* and perhaps identify treatments that have important implications for lung development in premature babies.

Specific points to address:

1. Expression of SOX9 and ID2 were analyzed to identify the distal tip progenitor population. SOX9 +ve progenitors were initially observed at the tips on day 0, became widespread by day 1 and localized to discrete patches on day 2 before becoming once again restricted to the distal tips of the epithelial branches on day 6. The increased number of cells expressing SOX9 is supported by the increase in *Sox9* expression observed by QRT-PCR at day 2. It would be helpful to further expand on this to discuss the potential implications for the transient widespread expression of SOX9. Is this an artifact of culture? Is this associated with proliferation and growth of the organoid? Or is it that the cells placed in culture have a higher percentage of SOX9 +ve cells and only those at what becomes the equivalent of the distal tip retain SOX9 expression? Did ID2 show the same expansion of expression domains before becoming restricted as suggest to the increase in Id2 expression? Does the relative decrease in mRNA expression at day 6 reflect a more limited number of SOX9 +ve cells as compared to the percentage of SOX9 +ve cells in the distal tips cells in the E13.5 population?

2. In addition to the % of bipotential, AT1 and AT2 cells in Figure 2B, the absolute numbers for the cells that were counted in each population should be included either in the text or in the figure legend. There also seems to be an increase in the percentage of AT2 cells but it is not clear if this is a significant increase.

3. The authors state that the proportion of bipotential progenitors and AT1 and AT2 cells do not change in a 30-day culture. Addition of a figure showing this would be great.

*Reviewer #3:*

Gkatzis and colleagues have developed a method for the expansion and cell fate specification of fetal lung alveolar progenitor cells from the developing mouse lung. Their refined methods focus on a serum free approach and they find that culture in collagen I promotes the differentiation to alveolar type 1 (AT1) cells with a typical flattened morphology. They also found, through a chemical screen, that Casein Kinase was a critical component of the media to maintain alveolar type 2 (AT2) cells and when it was inhibited, the AT2 cells differentiated to AT1 cells. The authors have successfully described a simple model system that can be used to characterize mechanisms regulating cellular stemness and cell fate decisions during fetal lung development. The data also provide and increased understanding of WNT signaling pathways in the support of distal airway specification. The paper provides a new model that has potential for screening of compounds regulating alveolar development and maturation.

The conclusions of the paper are supported by the data; however, it would benefit from some functional validation, increased characterization and further evaluation and mechanistic follow through on the chemical screen.

Strengths of the manuscript

The manuscript presents solid data evaluating the growth conditions essential to stimulate fetal organoid budding and branching elongation that maps nicely to E18.5 of fetal mouse development. They show good images of AT1 and AT2 cell differentiation and evidence of more primitive bipotent progenitors expressing both RAGE and SFTPC. To confirm maturity, nice apical and basolateral localization of alveolar cell markers including podoplanin, LAMP3, RAGE and ZO1 in differentiated AT1 cells. The authors demonstrate the utility of the model by performing a chemical screen of WNT modulators identifying compounds that increase the bipotent progenitors and compounds that increase AT1 cell specification.

Major points to consider:

It is unclear whether by day 6 of ex vivo growth the organoids stop growing and have the branches stopped elongating. If they are still growing, would it not be expected to still observe all the progenitor markers at the tips of the branches? It would be interesting to know what is restricting further growth of the branches and whetehr the progenitor cells are exhausted.

Only a couple of the markers used for characterization are discussed. While the majority of these genes follow a pattern similar to E18.5, it is notable that ATF3 is considerably higher in day 6 organoids than in E18.5 cells. As ATF3 is a transcription factor that can regulate many downstream pathways, such as epithelial mesenchymal transition and is itself regulated by oxidative stress – is its upregulation a result of artificial stress induced by the organoid system and is its higher expression impacting extended function of these cells?

There is a lack of data on the functional properties of the cells at Day 6. If these are to be considered mature, it would interesting to know whether they are capable of surfactant secretion, are lamellar bodies present.

It is interesting that the authors chose to evaluate the wnt modulators on day 6 of their experiment and not before if they were expecting them to impact mature cell formation, which they preciously describe as being at Day 6 with no information on day 8 cultures provided. Some additional rationale as to the choice of timepoint would be helpful to the flow of the paper.

While the strengths of the manuscript suggest the value of the model in understanding alveolar development, there are a number of points that should be considered in strengthening the impact and robustness of the manuscript.

In addition to the suggestions above the authors should consider the following additional data:

The characterization of the WNT signaling is restricted to a small amount of RNA analysis. Can the cultures be pushed toward mature AT2 cells? Can the cells grown in the presence of the modulators that increase the number of BP cells then re differentiate when the modulator is removed? Are the BP cells more proliferative? This would be interesting to study in the context of modulating an airways ability of proliferate and repair.

How important are the mesenchymal cells in this differentiation? At the beginning of the manuscript the authors state that the tips have mesenchymal and epithelial cells – what happened to the mesenchymal cells and how are they distributed in the day 6 organoids?

It would be nice to see some colocalization of the markers in panel 1E in addition to the SOX9 staining shown in Panel D.

As for Figure 1 it would be nice to see more extensive co-staining of the cells such as HTII-280 and SFTPC colocalization of AT2 cells and AQP5, RAGE, HOPX on the AT1 cells. EM images showing the presence of lamellar bodies and westerns for the presence of Pro surfactant B and C and SPB and SPC would be a nice addition as proof of functional differentiation.

In figure 4A it would be nice to see a wider panel of markers for AT1, AT2 and WNT signaling.

The characterization of the WNT signaling is a little disappointing and restricted to a small amount of RNA analysis. It would be nice to validate the data by in situ hybridization looking at co-localization between Axin2 and AT2/1/BP cells.

[Editors' note: further revisions were suggested prior to acceptance, as described below.]

Thank you for submitting your article "Differentiation of mouse fetal lung alveolar progenitors in serum-free organotypic cultures" for consideration by *eLife*. Your article has been reviewed by three peer reviewers, and the evaluation has been overseen by a Reviewing Editor and Edward Morrisey as the Senior Editor. The following individuals involved in review of your submission have agreed to reveal their identity: Aimee K Ryan (Reviewer #2); Amy Firth (Reviewer #3).

The manuscript has improved significantly and the editor and reviewers alike believe this study is novel and impactful. Reviewer #3 points out some of the limitations of the study, which we would like to ask you to explicitly address in the discussion part of the manuscript, for example in form of a paragraph focused on the limitations. These include potential influence of residual supportive mesenchymal cells as well as potential limitations of the wet screen including missing validation studies, limited concentrations used and potential heterogeneity of cells in the organoid assay.

No additional experiments are needed.

*Reviewer #1:*

The authors have addressed my critiques to satisfaction. This is an important study and will be of interest to the readership of *eLife*.

*Reviewer #2:*

I am satisfied with the authors' responses to my previous comments and have nothing additional to add.

*Reviewer #3:*

The authors have addressed several of my concerns and have overall improved the quality of the manuscript. I do still have some concerns relating to my initial comments that still dampen my enthusiasm for the impact of the data presented.

Overall, however, the model does offer a platform for potentially evaluating signaling mechanisms controlling alveolar fate decisions which would be of value to the community.

In response to my concern over the influence of the mesenchyme in alveolar specification and the potential influence of residual supportive mesenchymal cells, the authors acknowledge that organoid morphogenesis did not appear to depend on their prevalence, however there is no data to support this speculation. The amount of mesenchyme present could be vastly different per organoid and have significant impact on the data outcomes. Relating to this, the lack of impact of many of the wnt signaling regulators on the differentiation is interesting based on current data in the field. Unfortunately, the impact of the wnt screening is still hard to interpret as there is a lack of validation of the compounds on multiple biological and experimental replicates.

There is also no proof that the inhibitors, at the concentrations used for the screening, were optimal or actively inhibiting wnt signaling in this particular assay (a western to show the level of inhibition would be supportive). The rationale for the different CK2 inhibitors not having the same impact, is described as being due to their potential differences in potency/target affinity – why not perform a dose response curve?

Finally, the screen is presented as the % of alveolar cells, this population is likely vastly different in each organoid at the start of the assay and thus may have a significant impact on the outcomes. This should at least be described in some of the limitations of the assay and may lead to some regulators being overlooked.

---

## [Author Response]

Essential revisions:(1) Additional experimental data on Wnt readouts and Wnt/CK link

To strengthen our finding of Casein Kinase inhibitors downregulating the transcriptional response to WNT, we performed luciferase assays in HEK293T cells (Figure 4C) using the SuperTOPFlash reporter system (Veeman et al., 2003).

(2) Discussion on the potential role of mesenchymal/other non-epithelial cells in the organoid system.

To discuss the role of interstitial cells, we added a paragraph (page 3) stating that a possible contribution of mesenchymal cells to the growth and morphogenesis of organoids could not be excluded. Although mesenchymal cells with fibroblast morphology can be observed in our organoid cultures, in our chemical screen we did not observe any obvious correlation between the prevalence of fibroblasts and the manifestation of phenotypes.

(3) Discussion on the effect of Wnt inhibitors on cell fate specification

We have now included a paragraph discussing published data on the role of WNT signaling in alveolar cell fate specification (page 5). We refer to work showing that Axin2-positive AT2 cells differentiate into AT1 cells upon blockade of WNT transduction postnatally (Frank et al., 2016; Nabhan et al., 2018). We also mention similar work using hiPSC-derived AT2 cells and cultured human alveolar progenitors (Kanagaki et al., 2020 and Zacharias et al., 2018, respectively). To our knowledge, published data refer to postnatal alveolar progenitors, while the mechanisms underlying fetal progenitor differentiation remain to a large extent uncharacterized. Since Axin2positive AT2 progenitors have been described only in the context of postnatal alveologenesis, the fetal mechanisms of alveolar differentiation may differ at least in part. In addition to differences between the biological systems used in previous publications and our work, potential methodological limitations could also be part of the reasons for the observed discrepancy. We provide a discussion of these issues in the revised manuscript.

(4) Data/discussion on potential Sox2 differentiation

To assess whether cells in the organoids can express markers of proximal differentiation, we performed immunostaining for SOX2 and other proximal epithelial cell markers (FOXJ1, KRT5) at several stages of organoid development (Figure 1-Figure suppl. 1C, D). At all stages analyzed, we observed SOX2 expression specifically in the central portion of the organoids (core). In contrast to the distal SOX9^+^ region, the proximal *Sox2*^+^ domain did not appear to expand during culture. Although variable, FOXJ1 and KRT5 expression could be observed in sparse cells in the organoid core on day 6. This result suggests that, to some extent, proximal airway cells can be maintained or differentiate in our organoid model. We do not exclude the possibility that proximal airway cell composition could be enriched under different culture conditions. However, proximal cell marker expression was invariably observed to be localized in the central portion of the organoids, suggesting that distal epithelial branches provide a specific model for alveolar cell differentiation.

Reviewer #1:[…] 1. In this culture system, the authors found that distal progenitor cells could expand and become alveolar lineage epithelial cells (bipotent cells, AT1 and AT2). Considering that distal progenitor cells at E14.5 also could also differentiate into proximal airway lineage cells, the authors should provide information on whether these cells could differentiate into Sox2^+^ proximal progenitor cells and other proximal cell types.

We thank the Reviewer for this question. The treatment of explanted lung tissue with FGF7 and FGF10 promotes distal epithelial expansion (Bellusci et al., 1997), as observed in our model. To characterize proximal airway cell composition in our organoids, we performed immunostaining for SOX2, FOXJ1 and KRT5. This analysis revealed a proximal domain of SOX*2* expression in the organoid core, persisting at all culture stages analyzed (i.e., at least until day 6). In addition, FOXJ1 and KRT5, markers of ciliated and basal cells, respectively, were also variably expressed in sparse cells in the central portion of the organoids on day 6. Since proximal epithelial cells can be maintained and perhaps to some degree differentiate in our organoid model, we mention this point in the revised manuscript (page 3).

2. Based on their dissection method, the distal progenitor cells-derived organoids may contain mesenchymal cells or other non-epithelial cell types, which may significantly influence the proliferation and differentiation of distal progenitor epithelial cells. The authors should at least consider this possibility and discuss whether these drug-induced phenotypes are caused by modulation of mesenchymal cells or other non-epithelial cell types.

We thank the Reviewer for bringing up this important point. We now mention in the revised manuscript that mesenchymal cells are associated with the organoid epithelial branches and that a role for these cells in the morphogenesis of our model cannot be excluded. However, since we did not observe any clear correlation between the prevalence of mesenchymal cells and the phenotypes in our screen, we suggest that mesenchymal cells do not contribute significantly to the observed phenotypes. In addition, the reduction in WNT-dependent transcription observed in organoids (Figure 4B) could be recapitulated in cultured epithelial cells (HEK293T) expressing a WNT-reporter construct (Figure 4C), consistent with the model that the drugs are acting at least on epithelial cells (Figure 1-Figure suppl. 1B).

3. The electron microscopy analysis can better demonstrate the cell identity of bipotent cells, AT1 and AT2 cells in the organoids.

We thank the Reviewer for this suggestion. Although electron microscopy data would certainly help identify progenitors and differentiated cell types in the organoids, we have not yet established the required protocol.

4. Using this culture system to perform a Wnt signaling-related drug screening experiment, the authors found that Wnt inhibitors do not affect the cell fate specification, which does not agree with previous results. The authors should provide a discussion. Further details of this drug screening experiment should be provided in the manuscript.

We thank the Reviewer for this comment and have now addressed this issue by providing a discussion on WNT modulation of alveolar cell fate specification (page 5). Although WNT activity has been shown to control cell fate establishment in the context of postnatal development, we point out that the role of WNT signaling in alveolar differentiation at fetal stages is incompletely understood and may differ with what happens at postnatal stages. In addition, published work has probed the role of WNT signaling *in vivo* by genetic deletion or stabilization of *Ctnnb1*, which could result in distinct phenotypes compared with those caused by pharmacological inhibition of the pathway.

5. The link between Casein Kinase (CK) inhibition and Axin2 mRNA expression is not strong enough. The authors should provide more evidence to support their observed relationship between CK and Wnt signaling. Furthermore, the authors should discuss and propose the mechanisms of CK in regulating alveolar epithelial cell fate specification.

We thank the Reviewer for these comments and suggestions. To support our model that CK inhibition can lead to inhibition of β-catenin-dependent transcription, we performed SuperTOPflash-based luciferase assays and determined the effect of CK inhibitor compounds in HEK293T cells (Figure 4C). These experiments showed that IC261, CX-4945 and DMAT induced a significant decrease in WNT-dependent transcription.

Regarding the mechanisms of CK in regulating alveolar epithelial cell fate specification, we now mention in the revised manuscript: “[…] our results suggest that Casein Kinases antagonize cell differentiation by modulating WNT signaling transduction and transcriptional regulation […]” (page 6).

Reviewer #2:[…] Specific points to address:1. Expression of SOX9 and ID2 were analyzed to identify the distal tip progenitor population. SOX9 +ve progenitors were initially observed at the tips on day 0, became widespread by day 1 and localized to discrete patches on day 2 before becoming once again restricted to the distal tips of the epithelial branches on day 6. The increased number of cells expressing SOX9 is supported by the increase in Sox9 expression observed by QRT-PCR at day 2. It would be helpful to further expand on this to discuss the potential implications for the transient widespread expression of SOX9. Is this an artifact of culture? Is this associated with proliferation and growth of the organoid? Or is it that the cells placed in culture have a higher percentage of SOX9 +ve cells and only those at what becomes the equivalent of the distal tip retain SOX9 expression?

We thank the Reviewer for these questions. We have now analyzed in more detail the variation in SOX9 expression between days 0, 1, 2 and 6 of organoid culture and included these data in Figure 1D. To investigate the possible correlation with the proliferation of cultured progenitors, we have also now included confocal images showing co-expression of SOX9 and KI67. The transient widespread increase in SOX9 expression during the first days of culture closely follows cell proliferation dynamics in the organoids. By day 6, the reduction in SOX9 expression is accompanied by a reduced number of proliferating cells in the distal portion of the organoids, providing independent evidence that cell cycle exit and cell differentiation are likely occurring by this stage.

Did ID2 show the same expansion of expression domains before becoming restricted as suggest to the increase in Id2 expression? Does the relative decrease in mRNA expression at day 6 reflect a more limited number of SOX9 +ve cells as compared to the percentage of SOX9 +ve cells in the distal tips cells in the E13.5 population?

Unfortunately, ID2 expression could not be clearly analyzed in all samples due to technical issues with the mouse monoclonal antibody used, and thus we have now removed the ID2 data from the manuscript.

The revised manuscript now includes clearer information regarding the prevalence of SOX9^+^ cells at day 6 (new data included in Figure 1D). Indeed, as pointed out by the Reviewer, a reduction in the number of SOX9+ cells is apparent on day 6 compared to day 0 (E14.5), which correlates with the observed decrease in mRNA levels (Figure 1E).

2. In addition to the % of bipotential, AT1 and AT2 cells in Figure 2B, the absolute numbers for the cells that were counted in each population should be included either in the text or in the figure legend. There also seems to be an increase in the percentage of AT2 cells but it is not clear if this is a significant increase.

We thank the Reviewer for this comment and have now included absolute cell counts as supporting data (Figure 2-source data.xlsx and Figure 3-source data.xlsx). Unfortunately, the number of AT2 cells in our organoids is too low, making it difficult to determine the significance of potential increases in AT2 cell percentages.

3. The authors state that the proportion of bipotential progenitors and AT1 and AT2 cells do not change in a 30-day culture. Addition of a figure showing this would be great.

We thank the Reviewer for the suggestion. We believe that additional work would be needed to solidify our claim. Since the strength of our model lies in the rapid maturation and cell differentiation observed, we think that more information on the prolonged maintenance of organoids will not benefit the conclusions of this manuscript. For this reason, we have removed our original claim about long-term organoid cultures.

Reviewer #3:[…] Major points to consider:It is unclear whether by day 6 of ex vivo growth the organoids stop growing and have the branches stopped elongating. If they are still growing, would it not be expected to still observe all the progenitor markers at the tips of the branches? It would be interesting to know what is restricting further growth of the branches and whether the progenitor cells are exhausted.

We thank the Reviewer for these questions. To assess cell proliferation in the organoids, we performed KI67 immunostaining at several stages and included the data in Figure 1D. This analysis confirmed that whereas most cells are proliferating during the first two days of organoid culture, by day 6 the number of KI67+ cells has decreased substantially. This observation suggests that organoids are only slowly growing at this stage, although distal branches further elongate until day 8. In terms of progenitor marker expression at the tips of the branches, we observed SOX9 expression confined to a small subset of distal cells at day 6 (Figure 1D).

Only a couple of the markers used for characterization are discussed. While the majority of these genes follow a pattern similar to E18.5, it is notable that ATF3 is considerably higher in day 6 organoids than in E18.5 cells. As ATF3 is a transcription factor that can regulate many downstream pathways, such as epithelial mesenchymal transition and is itself regulated by oxidative stress – is its upregulation a result of artificial stress induced by the organoid system and is its higher expression impacting extended function of these cells?

We thank the Reviewer for these questions. We included *Atf3* in our analysis as a potential readout of the cellular stress response. It is possible that higher *Atf3* levels simply reflect transient oxidative or nutritional stress preceding medium change on day 6. Additional work will be required to localize *Atf3* expression in the organoids and determine whether the cellular stress response plays a role in cell differentiation.

There is a lack of data on the functional properties of the cells at Day 6. If these are to be considered mature, it would interesting to know whether they are capable of surfactant secretion, are lamellar bodies present.

We thank the Reviewer for these questions. We fully agree that a more detailed characterization of the cellular features of this system would be very interesting; however, a thorough analysis is preferable and will take some time to complete.

It is interesting that the authors chose to evaluate the wnt modulators on day 6 of their experiment and not before if they were expecting them to impact mature cell formation, which they preciously describe as being at Day 6 with no information on day 8 cultures provided. Some additional rationale as to the choice of timepoint would be helpful to the flow of the paper.

We thank the Reviewer for this comment. On day 6, approximately equal numbers of RAGE^+^/SFTPC^-^ cells and double positive cells are found in distal epithelial branches. This cellular composition evolved during the next two days and in the absence of growth factors into a significant increase in AT1 cell number at the expense of progenitors (see Author response image 1). Day 6 was therefore selected as an ideal time point for our chemical screen, since differentiation is ongoing at this stage. We have now included this important information in the revised manuscript (page 5).

**Author response image 1. sa2fig1:** 

While the strengths of the manuscript suggest the value of the model in understanding alveolar development, there are a number of points that should be considered in strengthening the impact and robustness of the manuscript.In addition to the suggestions above the authors should consider the following additional data:The characterization of the WNT signaling is restricted to a small amount of RNA analysis. Can the cultures be pushed toward mature AT2 cells? Can the cells grown in the presence of the modulators that increase the number of BP cells then re differentiate when the modulator is removed? Are the BP cells more proliferative? This would be interesting to study in the context of modulating an airways ability of proliferate and repair.

We thank the Reviewer for these very interesting questions which will definitely be worth addressing in the future.

How important are the mesenchymal cells in this differentiation? At the beginning of the manuscript the authors state that the tips have mesenchymal and epithelial cells – what happened to the mesenchymal cells and how are they distributed in the day 6 organoids?

We thank the Reviewer for these questions. By day 6 mesenchymal cells can be observed between epithelial branches (see Author response image 2, arrowheads point to mesenchymal cells); however, we did not observe a correlation between the number of mesenchymal cells and organoid development/morphology. We have now included an explanation on the possible role of mesenchymal cell in our system in the revised manuscript (page 3).

It would be nice to see some colocalization of the markers in panel 1E in addition to the SOX9 staining shown in Panel D.

We thank the Reviewer for this suggestion. We could not find working antibodies for most of the targets reported in panel 1E. However, we have now included a confocal series detailing the different expression domains of SOX9 and SOX2 in organoids at different stages (Figure 1-Figure suppl. 1C), which suggest that the organoids are patterned in the proximo-distal axis.

As for Figure 1 it would be nice to see more extensive co-staining of the cells such as HTII-280 and SFTPC colocalization of AT2 cells and AQP5, RAGE, HOPX on the AT1 cells. EM images showing the presence of lamellar bodies and westerns for the presence of Pro surfactant B and C and SPB and SPC would be a nice addition as proof of functional differentiation.

We thank the Reviewer for these suggestions. As noted above, further analysis of cell differentiation and maturation in the organoids will be most valuable.

In figure 4A it would be nice to see a wider panel of markers for AT1, AT2 and WNT signaling.The characterization of the WNT signaling is a little disappointing and restricted to a small amount of RNA analysis. It would be nice to validate the data by in situ hybridization looking at co-localization between Axin2 and AT2/1/BP cells.

We thank the Reviewer for these comments. Although we have not yet developed robust *in situ* hybridization techniques in our organoid model, we have validated the effect of CK inhibitors on WNT signaling by using a SuperTOPFlash luciferase reporter of WNT/β-catenin transcriptional activity (Figure 4C), which we hope will in part address the Reviewer’s concerns.

[Editors' note: further revisions were suggested prior to acceptance, as described below.]

The manuscript has improved significantly and the editor and reviewers alike believe this study is novel and impactful. Reviewer #3 points out some of the limitations of the study, which we would like to ask you to explicitly address in the discussion part of the manuscript, for example in form of a paragraph focused on the limitations. These include potential influence of residual supportive mesenchymal cells as well as potential limitations of the wet screen including missing validation studies, limited concentrations used and potential heterogeneity of cells in the organoid assay.No additional experiments are needed.

We thank the Editors and Reviewers for their supportive comments and helpful suggestions. We have now revised the manuscript to include a paragraph discussing the limitations of our screening approach, which should facilitate the interpretation of the presented data.

Reviewer #3:The authors have addressed several of my concerns and have overall improved the quality of the manuscript. I do still have some concerns relating to my initial comments that still dampen my enthusiasm for the impact of the data presented.Overall, however, the model does offer a platform for potentially evaluating signaling mechanisms controlling alveolar fate decisions which would be of value to the community.In response to my concern over the influence of the mesenchyme in alveolar specification and the potential influence of residual supportive mesenchymal cells, the authors acknowledge that organoid morphogenesis did not appear to depend on their prevalence, however there is no data to support this speculation. The amount of mesenchyme present could be vastly different per organoid and have significant impact on the data outcomes. Relating to this, the lack of impact of many of the wnt signaling regulators on the differentiation is interesting based on current data in the field. Unfortunately, the impact of the wnt screening is still hard to interpret as there is a lack of validation of the compounds on multiple biological and experimental replicates.There is also no proof that the inhibitors, at the concentrations used for the screening, were optimal or actively inhibiting wnt signaling in this particular assay (a western to show the level of inhibition would be supportive). The rationale for the different CK2 inhibitors not having the same impact, is described as being due to their potential differences in potency/target affinity – why not perform a dose response curve?Finally, the screen is presented as the % of alveolar cells, this population is likely vastly different in each organoid at the start of the assay and thus may have a significant impact on the outcomes. This should at least be described in some of the limitations of the assay and may lead to some regulators being overlooked.

We thank the Reviewer for their helpful suggestions. In the revised manuscript, we have included a paragraph outlining the limitations of our screen. We agree that the variable mesenchymal cell content in the organoids and the single concentration used for the compounds could have influenced the results. And while the screened compounds were not pre-validated, the screen was performed in four biological replicates (different pregnant dams), which allowed us to identify compounds that significantly shifted the differentiation of alveolar progenitors. This important information is included in the legend of Figure 3 and we have now also included it in the Methods section, under Chemical screen. We further show that while the relative composition of alveolar cells at the start of the assay (day 6) is variable, it is not vastly different among different replicates (Figure 2B).